# Generalized Michaelis–Menten rate law with time-varying molecular concentrations

**Roktaek Lim**[1,2☯], **Thomas L. P. Martin**[1☯], **Junghun Chae**[2☯], **Woo Joong Kim**[2], **Cheol-Min Ghim**[2,3]*, **Pan-Jun Kim**[1,4,5,6]*

**1** Department of Biology, Hong Kong Baptist University, Kowloon, Hong Kong, **2** Department of Physics, Ulsan National Institute of Science and Technology, Ulsan, Republic of Korea, **3** Department of Biomedical Engineering, Ulsan National Institute of Science and Technology, Ulsan, Republic of Korea, **4** Center for Quantitative Systems Biology & Institute of Computational and Theoretical Studies, Hong Kong Baptist University, Kowloon, Hong Kong, **5** State Key Laboratory of Environmental and Biological Analysis, Hong Kong Baptist University, Kowloon, Hong Kong, **6** Abdus Salam International Centre for Theoretical Physics, Trieste, Italy

☯ These authors contributed equally to this work.
* cmghim@unist.ac.kr (C-MG); extutor@gmail.com (P-JK)

**Data Availability Statement:** All relevant data are provided in Supporting information. Source codes are available at https://github.com/rokt-lim/Generalized_Michaelis-Menten_rate_law.

## Abstract

The Michaelis–Menten (MM) rate law has been the dominant paradigm of modeling biochemical rate processes for over a century with applications in biochemistry, biophysics, cell biology, systems biology, and chemical engineering. The MM rate law and its remedied form stand on the assumption that the concentration of the complex of interacting molecules, at each moment, approaches an equilibrium (quasi-steady state) much faster than the molecular concentrations change. Yet, this assumption is not always justified. Here, we relax this quasi-steady state requirement and propose the generalized MM rate law for the interactions of molecules with active concentration changes over time. Our approach for time-varying molecular concentrations, termed the effective time-delay scheme (ETS), is based on rigorously estimated time-delay effects in molecular complex formation. With particularly marked improvements in protein–protein and protein–DNA interaction modeling, the ETS provides an analytical framework to interpret and predict rich transient or rhythmic dynamics (such as autogenously-regulated cellular adaptation and circadian protein turnover), which goes beyond the quasi-steady state assumption.

## Author summary

The Michaelis–Menten (MM) rate law has enjoyed for over a century the status of the de facto standard of modeling enzymatic reactions. Despite its simple and intuitive interpretation for a wide range of applications in biochemistry, biophysics, cell biology, systems biology, and chemical engineering, the MM rate law and its modified form stand on the quasi-steady state assumption, which is not necessarily justified under active molecular concentration changes over time. Here, we relax this assumption and propose the generalized MM rate law where the effective time delay in molecular complex formation comes into pivotal play. This scheme allows the analytical interpretation and prediction of

**Funding:** This work was supported by Hong Kong Baptist University, Startup Grant Tier 2 (RC-SGT2/18-19/SCI/001) and Blue Sky Research Fund (RC-BSRF/21-22/09) (R.L., T.L.P.M., and P.-J.K.), the Health and Medical Research Fund (HMRF 17182691) (R.L. and P.-J.K.), and the National Research Foundation of Korea Grants (NRF-2020R1A4A101914013, NRF-2020R1F1A107594213, and NRF-2018K1A4A3A01063890) funded by the Ministry of Science and ICT (J.C., W.J.K., and C.-M.G). The funders had no role in study design, data collection and analysis, decision to publish, or preparation of the manuscript.

**Competing interests:** The authors have declared that no competing interests exist.

various biochemical processes with transient or rhythmic dynamics, opening a new avenue of applications beyond the previous approaches.

## Introduction

Since proposed by Henri [1] and Michaelis and Menten [2], the Michaelis–Menten (MM) rate law has been the dominant framework for modeling the rates of enzyme-catalyzed reactions for over a century [1–4]. The MM rate law has also been widely adopted for describing other bimolecular interactions, such as reversible binding between proteins [5–7], between a gene and a transcription factor [8,9], and between a receptor and a ligand [10,11]. The MM rate law hence serves as a common mathematical tool in both basic and applied fields, including biochemistry, biophysics, pharmacology, systems biology, and many subfields of chemical engineering [12]. The derivation of the MM rate law from the underlying biochemical mechanism is based on the steady-state approximation by Briggs and Haldane [3], referred to as the *standard quasi-steady state approximation* (sQSSA) [12–17]. The sQSSA, however, is only valid when the enzyme concentration is low enough and thus the concentration of enzyme–substrate complex is negligible compared to substrate concentration [14]. This condition may be acceptable for many metabolic reactions with substrate concentrations that are typically far higher than the enzyme concentrations.

Nevertheless, in the case of protein–protein interactions in various cellular activities, the interacting proteins as the "enzymes" and "substrates" often show the concentrations comparable with each other [18–20]. Therefore, the use of the MM rate law for describing protein–protein interactions has been challenged in its rationale, with the modified alternative formula from the *total quasi-steady state approximation* (tQSSA) [12,13,17,21–27]. The tQSSA-based form is generally more accurate than the MM rate law from the sQSSA, for a broad range of combined molecular concentrations and thus for protein–protein interactions as well [12,13,21–27]. The superiority of the tQSSA has not only been proven in the quantitative, but also in the qualitative outcomes of systems, which the sQSSA sometimes fails to predict [12,21]. Later, we will provide the overview of the tQSSA and its relationship with the conventional MM rate law from the sQSSA.

Despite the correction of the MM rate law by the tQSSA, both the tQSSA and sQSSA still rely on the assumption that the concentration of the complex of interacting molecules, at each moment, approaches an equilibrium (quasi-steady state) much faster than the molecular concentrations change [12,14,24]. Although this quasi-steady state assumption may work for a range of biochemical systems, the exact extent of such systems to follow that assumption is not clear. Numerous cellular processes do exhibit active molecular concentration changes over time, such as in signal responses, circadian oscillations, and cell cycles [6,7,21,28–31], calling for a better approach to even cover the time-varying molecular concentrations that may not strictly adhere to the quasi-steady state assumption.

In this study, we report the generalization of the MM rate law, whereby the interaction of time-varying molecular components is more properly described than by the tQSSA and sQSSA. This generalization is the correction of the tQSSA with rigorously estimated, time-delay effects affected by free molecule availability. Our formulation, termed the effective time-delay scheme (ETS), well accounts for the transient or oscillatory dynamics and experimental data patterns of biochemical systems with the relevant analytical insights, which are not captured by the previous methods. Surprisingly, we reveal that the existing quasi-steady state assumption can even fail for extremely slow changes in protein concentrations under

autogenous regulation, whereas the ETS does not. In addition, the ETS allows the natural explanation of rhythmic degradation of circadian proteins without requiring explicitly-rhythmic post-translational mechanisms; this is not straightforward within the quasi-steady state assumption. As an added feat, the ETS improves kinetic parameter estimation. As demonstrated in a number of contexts such as autogenously-regulated cellular adaptation and circadian oscillations, our approach offers a useful theoretical framework to interpret and predict rich transient or rhythmic dynamics of biochemical systems with a wide range of applicability.

## Results

### Theory development

First, we present the outline of the tQSSA, sQSSA, and our generalized MM rate law. Consider two different molecules A and B that bind to each other and form complex AB, as illustrated in Fig 1(A). For example, A and B may represent two participant proteins in heterodimer formation, a chemical substrate and an enzyme in a metabolic reaction, and a solute and a transporter in membrane transport. The concentration of the complex AB at time $t$, denoted by $C(t)$, changes over time as in the following equation from the mass-action law:

$$\frac{\mathrm{d}C(t)}{\mathrm{d}t} = k_{\mathrm{a}}[A(t) - C(t)][B(t) - C(t)] - k_{\delta}C(t). \tag{1}$$

Here, $A(t)$ and $B(t)$ denote the total concentrations of A and B, respectively, and hence $A(t) - C(t)$ and $B(t) - C(t)$ are the concentrations of free A and B. The temporal profiles of $A(t)$ and $B(t)$ are allowed to be very generic, e.g., even with their own feedback effects as addressed later. $k_{\mathrm{a}}$ denotes the association rate of free A and B. $k_{\delta}$ is the effective "decay" rate of AB with $k_{\delta} \equiv k_{\mathrm{d}} + r_{\mathrm{c}} + k_{\mathrm{loc}} + k_{\mathrm{dlt}}$ where $k_{\mathrm{d}}$, $k_{\mathrm{loc}}$, and $k_{\mathrm{dlt}}$ stand for the dissociation, translocation, and dilution rates of AB, respectively, and $r_{\mathrm{c}}$ for the chemical conversion or translocation rate of A or B upon the formation of AB. In other words, for the sake of generality, $k_{\delta}$ is not limited to a dissociation event but encompasses all rate events to lower the level of AB [Fig 1(A)].

In the tQSSA, the assumption is that $C(t)$ approaches the quasi-steady state fast enough each time, given the values of $A(t)$ and $B(t)$ [12,24]. This assumption and the notation $K \equiv k_{\delta}/k_{\mathrm{a}}$ lead Eq (1) to an estimate $C(t) \approx C_{\mathrm{tQ}}(t)$ with the following form (Text A in S1 Appendix):

$$C_{\mathrm{tQ}}(t) \equiv \frac{1}{2}\left\{K + A(t) + B(t) - K\Delta_{\mathrm{tQ}}(t)\right\}, \tag{2}$$

$$\Delta_{\mathrm{tQ}}(t) \equiv \sqrt{\left[1 + \frac{A(t) + B(t)}{K}\right]^2 - \frac{4}{K^2}A(t)B(t)}$$

$$= \sqrt{1 + 2\left[\frac{A(t) + B(t)}{K}\right] + \left[\frac{A(t) - B(t)}{K}\right]^2}. \tag{3}$$

Although the tQSSA looks a little complex, it only involves a single parameter $K$ and is easy to implement in a computer program. As mentioned earlier, the tQSSA is generally more accurate than the conventional MM rate law [12,13,21–27]. To obtain the MM rate law, consider a rather specific condition,

$$B(t) \ll K + A(t) \quad \text{or} \quad A(t) \ll K + B(t). \tag{4}$$

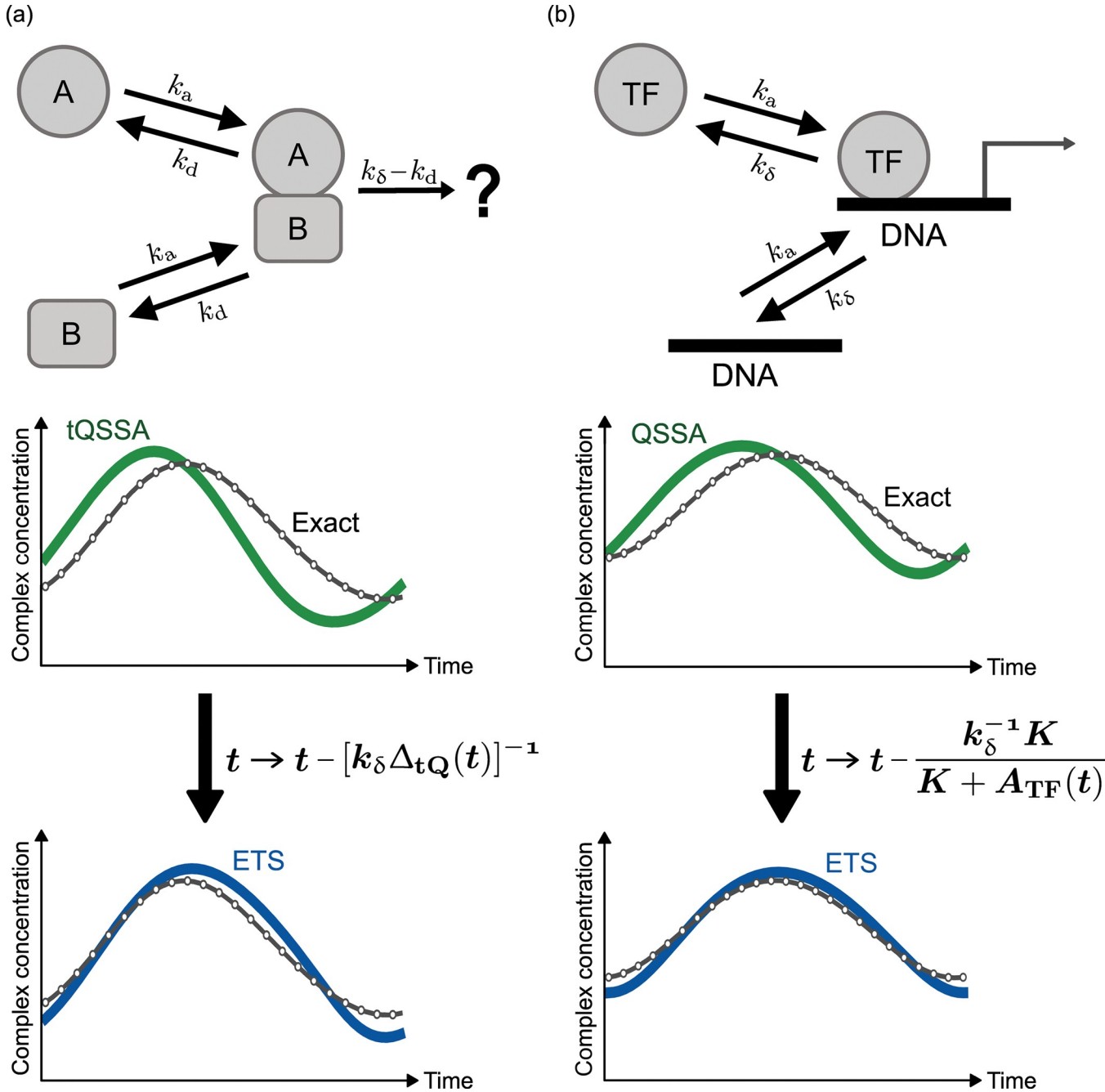

**Fig 1. Generalization of the MM rate law for time-varying molecular concentrations, referred to as the ETS.** (a) Two different molecules A and B bind to each other and form their complex. (b) A TF binds to a DNA molecule to regulate mRNA expression (RNA polymerase and other molecules are omitted here). In (a) and (b), the graphs show the comparison among the exact time-course profile of the complex concentration, the tQSSA-based (a) or QSSA-based (b) profile, and the ETS-based profile. The relationship between the tQSSA (or QSSA) and the ETS is illustrated through the effective time delay in the ETS. Notations $k_a$, $k_d$, $k_\delta$, $t$, $\Delta_{tQ}(t)$, $K$, and $A_{TF}(t)$ are defined in the description of Eqs (1)–(3) and (6)–(8). Simulations in (a) and (b) are based on periodic oscillation models in Texts G and H in S1 Appendix, respectively, with their parameters in Table G in S1 Appendix.

In this condition, the Padé approximant for $C_{tQ}(t)$ takes the following form:

$$C_{tQ}(t) \approx \frac{A(t)B(t)}{K + A(t) + B(t)}. \tag{5}$$

Considering Eq (5), Eq (4) is similar to the condition $C_{tQ}(t)/A(t) \ll 1$ or $C_{tQ}(t)/B(t) \ll 1$. In other words, Eq (5) would be valid when the concentration of AB complex is negligible compared to either A or B's concentration. This condition is essentially identical to the assumption in the sQSSA resulting in the MM rate law [14]. In the example of a typical metabolic reaction with $B(t) \ll A(t)$ for substrate A and enzyme B, Eq (4) is automatically satisfied and Eq (5) further reduces to the familiar MM rate law $C_{tQ}(t) \approx A(t)B(t)/[K + A(t)]$, i.e., the outcome of the sQSSA [1–4,12–14]. To be precise, the sQSSA uses the concentration of free A instead of $A(t)$, but we refer to this formula with $A(t)$ as the sQSSA because the complex is assumed to be negligible in that scheme. Clearly, $K$ here is the Michaelis constant, commonly known as $K_M$.

The application of the MM rate law beyond the condition in Eq (4) invites a risk of erroneous modeling results, whereas the tQSSA is relatively free of such errors and has wider applicability [12,13,21–27]. Still, both the tQSSA and sQSSA stand on the assumption that $C(t)$ approaches the quasi-steady state fast enough each time before the marked temporal change of $A(t)$ or $B(t)$. We now relax this quasi-steady state assumption and generalize the approximation of $C(t)$ to the case of time-varying $A(t)$ and $B(t)$, as the main objective of this study.

Suppose that $C(t)$ may not necessarily approach the quasi-steady state each time but stays within some distance from it. As detailed in Text A in S1 Appendix, we linearize the right-hand side of Eq (1) around $C(t) - C_{tQ}(t)$ and estimate $C(t)$'s solution as the time integral of $C_{tQ}(t')$ (where $t' \leq t$) with an exponential kernel-like function. The Taylor expansion of $C_{tQ}(t')$ by $t - t'$ is incorporated into this integral and then its form offers the following approximant for $C(t)$:

$$C_\gamma(t) \equiv \min\{C_{tQ}\{t - [k_\delta \Delta_{tQ}(t)]^{-1}\}, A(t), B(t)\}. \tag{6}$$

Although the above $C_\gamma(t)$ looks rather complex, this form is essentially a simple conversion $t \to t - [k_\delta \Delta_{tQ}(t)]^{-1}$ in the tQSSA. min$\{\cdot\}$ is just taken for a minor role to ensure that the complex concentration cannot exceed $A(t)$ or $B(t)$. Hence, the distinct feature of $C_\gamma(t)$ is the inclusion of an effective time delay $[k_\delta \Delta_{tQ}(t)]^{-1}$ in complex formation. This delay is the rigorous estimate of the molecular relaxation time during which the effect of instantaneous $A(t)$ and $B(t)$ is notably sustained in the complex formation, as shown in Text A in S1 Appendix. We will refer to this formulation as the effective time-delay scheme (ETS), and its relationship with the tQSSA is depicted in Fig 1(A).

We propose the ETS as the generalization of the MM rate law for time-varying molecular concentrations that may not strictly adhere to the quasi-steady state assumption. If the relaxation time in complex formation is so short that the effective time delay in Eq (6) can be ignored, the ETS returns to the tQSSA in its form. Surprisingly, we proved that, unlike the ETS, any simpler new rate law without a time-delay term would not properly work for active concentration changes over time (Text C in S1 Appendix). Nevertheless, one may question the analytical utility of the ETS, regarding the apparent complexity of its time-delay term. In the examples of autogenously-regulated cellular adaptation and rhythmic protein turnover below, we will use the ETS to deliver valuable analytical insights into the systems whose dynamics is otherwise ill-explained by the conventional approaches.

About the physical interpretation of the ETS, we notice that the effective time delay is inversely linked to free molecule availability, as $[k_\delta \Delta_{tQ}(t)]^{-1} =$

$k_\delta^{-1}\{1 + K^{-1}[A(t) + B(t) - 2C_{tQ}(t)]\}^{-1}$ from Eq (2). Here,
$A(t) + B(t) - 2C_{tQ}(t) = [A(t) - C_{tQ}(t)] + [B(t) - C_{tQ}(t)]$, which is the total free molecule concentration at the quasi-steady state each time. In other words, the less the free molecules, the more the time delay, which is at most $k_\delta^{-1}$. One can understand this observation as follows: $-k_\delta C(t)$ in Eq (1) gives the expectation that the decay time-scale ($k_\delta^{-1}$) of the complex may approximate the relaxation time. Yet, the relaxation time is shorter than $k_\delta^{-1}$, because free A and B are getting depleted over time as a result of their complex formation and therefore the complex formation rate $k_a[A(t) - C(t)][B(t) - C(t)]$ in Eq (1) continues to decline towards quicker relaxation of the complex level. This free-molecule depletion effect to shorten the relaxation time is roughly proportional to the free molecule concentration itself (Text A in S1 Appendix). Hence, the relaxation time takes a decreasing function of the free molecule concentration, consistent with the above observation. Clearly, the free molecule concentration would be low for relatively few A and B molecules with comparable concentrations—i.e., small $A(t)+B(t)$ and $[A(t)-B(t)]^2$ in Eq (3). In this case, the relaxation time would be relatively long and the ETS shall be deployed instead of the tQSSA or sQSSA. We thus expect that protein–protein interactions would often be the cases in need of the ETS compared to metabolic reactions with much excess substrates not binding to enzymes, as will be shown later.

Thus far, we have implicitly assumed the continuous nature of molecular concentrations as in Eq (1). However, there exist biomolecular events that fundamentally deviate from this assumption. For example, a transcription factor (TF) binds to a DNA molecule in the nucleus to regulate mRNA expression and the number of such a TF–DNA assembly would be either 1 or 0 for a DNA site that can afford at most one copy of the TF [Fig 1(B)]. This inherently discrete and stochastic nature of the TF–DNA assembly is seemingly contrasted with the continuous and deterministic nature of the molecular complex in Eq (1). To rigorously describe this TF–DNA binding dynamics, we harness the chemical master equation [32] and introduce quantities $A_{TF}(t)$ and $C_{TF}(t)$, which are the total TF concentration and the TF–DNA assembly concentration averaged over the cell population, respectively (Text D in S1 Appendix). According to our calculation, the quasi-steady state assumption leads to the following approximant for $C_{TF}(t)$:

$$C_{TFQ}(t) \equiv \frac{A_{TF}(t)}{V[K + A_{TF}(t)]}, \tag{7}$$

where $K \equiv k_\delta/k_a$ with $k_a$ and $k_\delta$ as the TF–DNA binding and unbinding rates, respectively, and $V$ is the nuclear volume (Text D in S1 Appendix). In fact, $C_{TFQ}(t)$ in Eq (7) corresponds to a special case of the previously-studied, *stochastic quasi-steady state approximation* (stochastic QSSA) [33,34] for arbitrary molecular copy numbers such as for multiple DNA binding sites. Of note, the stochastic QSSA becomes close to the tQSSA as its deterministic version, if $VK\Delta_{tQ}(t) \gg 1$ [34].

$C_{TFQ}(t)$ in Eq (7) looks very similar to the MM rate law, considering the "concentration" of the DNA site ($V^{-1}$). Nevertheless, $C_{TFQ}(t)$ is not a mere continuum of Eq (5), because the denominator in $C_{TFQ}(t)$ includes $K+A_{TF}(t)$, but not $K+A_{TF}(t)+V^{-1}$. In fact, the discrepancy between $C_{TFQ}(t)$ and Eq (5) comes from the inherent stochasticity in the TF–DNA assembly (Text D in S1 Appendix). In this regard, directly relevant to $C_{TFQ}(t)$ is the stochastic version of the MM rate law with denominator $K+A(t)+B(t)-V^{-1}$ proposed by Levine and Hwa [35], because the DNA concentration $B(t)$ is $V^{-1}$ in our case. $C_{TFQ}(t)$ is a fundamentally more correct approximant for the DNA-binding TF level than both the tQSSA and sQSSA in Eqs (2) and (5). Therefore, we will just refer to $C_{TFQ}(t)$ as the QSSA for the TF–DNA interactions.

Still, the use of $C_{\text{TFQ}}(t)$ stands on the quasi-steady state assumption. We relax this assumption and generalize the approximation of $C_{\text{TF}}(t)$ to the case of time-varying TF concentration. In a similar way to obtain $C_\gamma(t)$ in Eq (6), we propose the following approximant for $C_{\text{TF}}(t)$ (Text D in S1 Appendix):

$$C_{\text{TF}\gamma}(t) \equiv C_{\text{TFQ}}\left[t - \frac{k_\delta^{-1}K}{K + A_{\text{TF}}(t)}\right]. \qquad (8)$$

This formula represents the TF–DNA version of the ETS, and its relationship with the QSSA is illustrated in Fig 1(B). The time-delay term in Eq (8) has a similar physical interpretation to that in Eq (6). Besides, this term is directly proportional to the probability of the DNA unoccupancy at the quasi-steady state, according to Eq (7).

Through numerical simulations of various systems with empirical data analyses, we found that the ETS provides the reasonably accurate description of the deviations of time-course molecular profiles from the quasi-steady states (Texts F–H in S1 Appendix). This result was particularly evident for the cases of protein–protein and TF–DNA interactions with time-varying protein concentrations. In these cases, the ETS unveils the importance of the relaxation time (effective time delay) in complex formation to the shaping of molecular profiles, otherwise difficult to clarify. Yet, the use of the sQSSA or tQSSA is practically enough for typical metabolic reaction and transport systems, without the need for the ETS. The strict mathematical conditions for the validity of the ETS as well as those of the quasi-steady state assumption are derived in Text E in S1 Appendix.

## Autogenous control

Adaptation to changing environments is a process of biological control. The ETS offers an analytical tool for understanding transient dynamics of such adaptation processes, exemplified by autogenously regulated systems where TFs regulate their own transcription. This autogenous control underlies cellular responses to various internal and external stimuli [36,37]. We here explore the case of positive autoregulation and show that the quasi-steady state assumption does not even work for extremely-slow protein changes near a tipping point. The case of negative autoregulation is covered in Text J in S1 Appendix.

In the case of positive autoregulation, consider a scenario in Fig 2(A) that proteins enhance their own transcription after homodimer formation and this dimer–promoter interaction is facilitated by inducer molecules. The inherent cooperativity from the dimerization is known to give a sigmoid TF–DNA binding curve, resulting in abrupt and history-dependent transition events [36,38]. We here built the full kinetic model of the system without the ETS, tQSSA, or other approximations of the dimerization and dimer–promoter interaction (Text I in S1 Appendix). As the simulated inducer level increases, Fig 2(B) demonstrates that an initially low, steady-state protein level undergoes an abrupt leap at some point $\eta_c$, known as a transition or tipping point. This discontinuous transition with only a slight inducer increase signifies a qualitative change in the protein expression state. Reducing the inducer level just back to the transition point $\eta_c$ does not reverse the protein state, which is sustained until more reduction in the inducer level [Fig 2(B)]. This history-dependent behavior, hysteresis, indicates the coexistence of two different stable states of the protein level (bistability) between the forward and backward transitions [36,38].

Other than steady states, we examine how fast the system responds to signals. Upon acute induction from a zero to certain inducer level ($>\eta_c$), the protein level grows over time towards its new steady state and this response becomes rapider at stronger induction away from the transition point [Fig 2(C)]. Conversely, as the inducer level decreases towards the transition

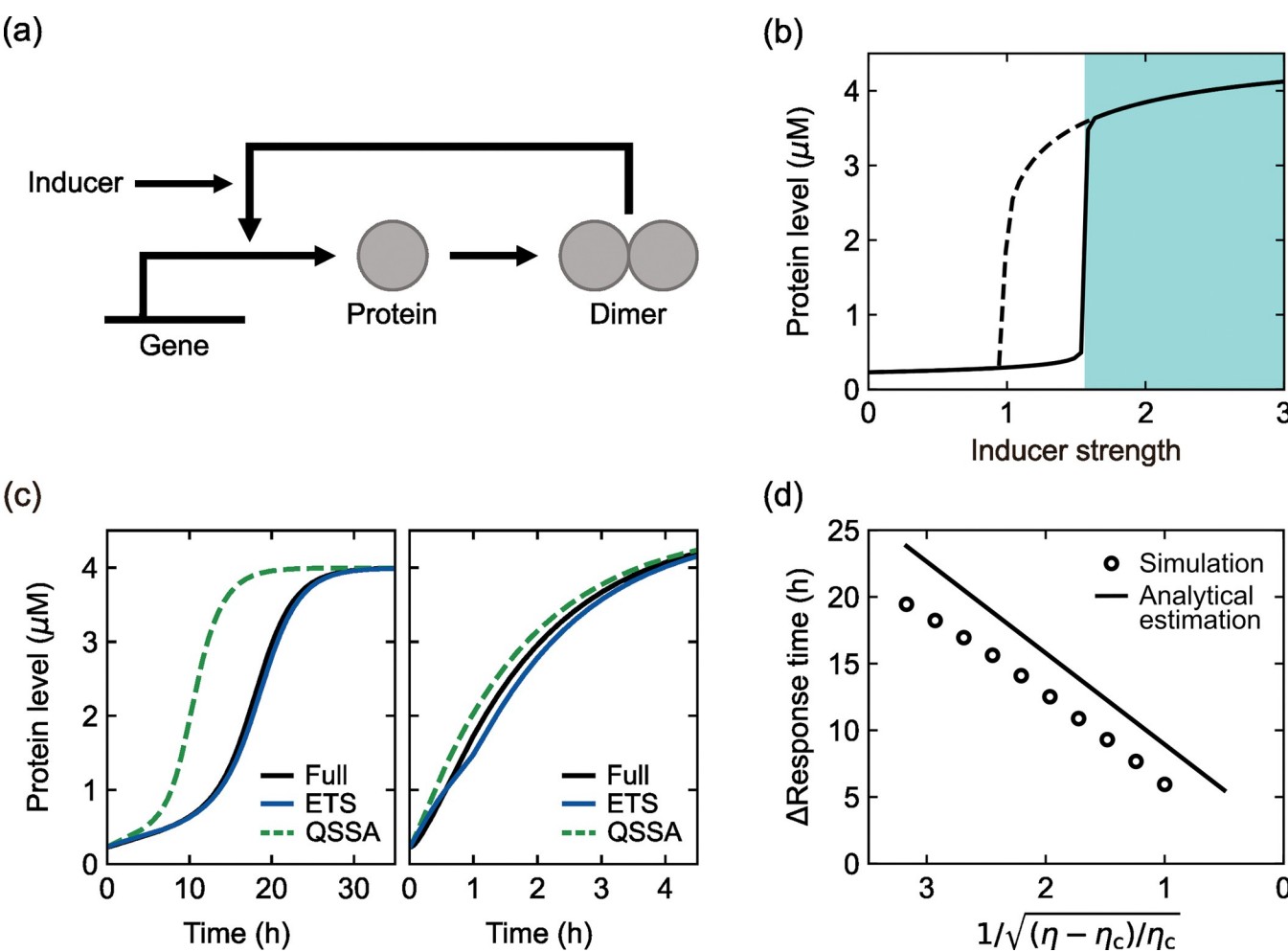

**Fig 2. Positive autoregulation and induction kinetics.** (a) Protein production mechanism with positive autoregulation in the presence of inducers. (b) Bifurcation diagram of the simulated protein level as a function of $\eta$ (proxy for an inducer level). The steady state is plotted as $\eta$ increases (solid line) or decreases (dashed line). Acute induction can be simulated by a sudden change of $\eta = 0$ to $\eta > \eta_c$ in the shaded area. (c) Time-series of protein levels from the full, ETS, and QSSA models upon acute induction at time 0 h with $\eta = 2.42$ (left) or $\eta = 200$ (right). (d) The full model-to-QSSA difference in response time as a function of $1/\sqrt{(\eta - \eta_c)/\eta_c}$. Both the simulated and analytically-estimated differences are presented. The analytical estimation is based on Eq (9). For more details of (b)–(d), refer to Text I and Tables H and I in S1 Appendix.

point, the response time continues to increase and eventually becomes diverging (in this study, response time is defined as the time taken for a protein level to reach 90% of its steady state). This phenomenon has been called "critical slowing down" [39–41]. Regarding this near-transition much slow protein growth, one may expect that the quasi-steady state assumption would work properly near that transition point. To test this possibility, we modified the full model by the tQSSA and QSSA of the dimerization and dimer-promoter interaction, respectively, and call this modified model the QSSA-based model. For comparison, we created another version of the model by the ETS of the dimerization and dimer-promoter interaction and call this version the ETS-based model (Text I in S1 Appendix). Across physiologically-relevant parameter conditions, we compared the QSSA- and ETS-based model simulation results to the full model's (Text L and Table E in S1 Appendix). Surprisingly, the QSSA-based model often severely underestimated the response time, particularly near a transition point, while the ETS-based response time was relatively close to that from the full model [$P < 10^{-4}$ and Text L in

S1 Appendix; e.g., 8.5-hour shorter and 0.5-hour longer response times in Fig 2(C) (left) in the QSSA and ETS cases, respectively].

This unexpected mismatch between the QSSA and full model results comes from the following factors: because the QSSA model discards the effective time delay in dimerization and dimer-promoter interaction, this model accelerates positive feedback, transcription, and protein production, and thus shortens the response time. Near the transition point, although the protein level grows very slowly, a little higher transcription activity in the QSSA model substantially advances the protein growth with near-transition ultrasensitivity that we indicated above. Therefore, the QSSA model shortens the response time even near the transition point.

Related to this point, the ETS allows the analytical calculation of response time and its QSSA-based estimate. In this calculation, we considered two different stages of protein growth —its early and late stages (Text I in S1 Appendix) and found that the QSSA model underestimates response time mainly at the early stage. This calculation suggests that the exact response time would be longer than the QSSA-based estimate by

$$
\frac{2\pi}{r\sqrt{\frac{\eta - \eta_{\mathrm{c}}}{\eta_{\mathrm{c}}}}} \left( \frac{1}{D} + \frac{1}{D_{\mathrm{TF}}} \right) + \frac{1}{r}\ln\left( 1 + \frac{1}{D} + \frac{1}{D_{\mathrm{TF}}} \right)
$$

$$
+ \frac{1}{r}\left( \frac{1}{D} + \frac{1}{D_{\mathrm{TF}}} \right)\ln\left\{ 1 + \frac{DD_{\mathrm{TF}}(\bar{u} - 1)[DD_{\mathrm{TF}}(\bar{u} - 1) - 2(D + D_{\mathrm{TF}})]}{(D + D_{\mathrm{TF}})^2} \right\}, \quad (9)
$$

where $\eta$ and $\eta_{\mathrm{c}}$ denote an inducer level and its value at the transition point, respectively, $r$ is the sum of protein degradation and dilution rates, and $D$ and $D_{\mathrm{TF}}$ are parameters inversely proportional to the effective time delays in dimerization and dimer–promoter interaction, respectively. The additional details and the definition of parameter $\bar{u}$ are provided in Text I in S1 Appendix.

Notably, the above response time difference vanishes as $D^{-1} + D_{\mathrm{TF}}^{-1} \to 0$. In other words, the total effective time delay is responsible for this response time difference. Strikingly, this difference indefinitely grows as $\eta$ decreases towards $\eta_{\mathrm{c}}$, as a linear function of $1/\sqrt{(\eta - \eta_{\mathrm{c}})/\eta_{\mathrm{c}}}$. This prediction can serve as a testbed for our theory and highlights far excessive elongation of near-transition response time (compared to the QSSA) as an amplified effect of the relaxation time in complex formation. This amplified effect is the result of the near-transition ultrasensitivity that we indicated above. Consistent with our prediction, the full model simulation always shows longer response time than the QSSA model simulation and the difference is linearly scaled to $1/\sqrt{(\eta - \eta_{\mathrm{c}})/\eta_{\mathrm{c}}}$ as exemplified by Fig 2(D) ($R^2 > 0.98$ in simulated conditions; see Text L in S1 Appendix). Moreover, its predicted slope against $1/\sqrt{(\eta - \eta_{\mathrm{c}})/\eta_{\mathrm{c}}}$ [i.e., $2\pi$ (6.28···) multiplied by $(D_{\mathrm{TF}}^{-1} + D^{-1})r^{-1}$] is comparable with the simulation results [7.3 ± 0.3 (avg. ± s.d. in simulated conditions) multiplied by $(D_{\mathrm{TF}}^{-1} + D^{-1})r^{-1}$; see Text L in S1 Appendix]. The agreement of these nontrivial predictions with the numerical simulation results proves the theoretical value of the ETS. Again, we raise a caution against the quasi-steady state assumption, which unexpectedly fails for very slow dynamics with severe underestimation of response time, e.g., by a few tens of hours in the case of Fig 2(D).

## Rhythmic degradation of circadian proteins

Circadian clocks in various organisms generate endogenous molecular oscillations with ~24 h periodicity, enabling physiological adaptation to diurnal environmental changes caused by the Earth's rotation around its axis. Circadian clocks play a pivotal role in maintaining biological homeostasis, and the disruption of their function is associated with a wide range of

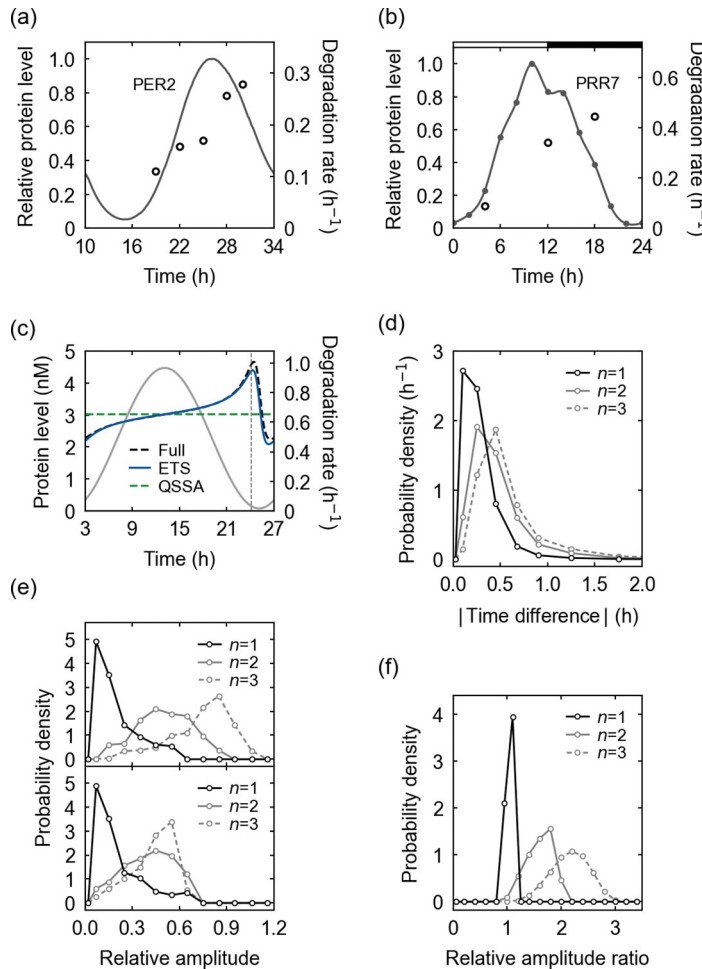

**Fig 3. Rhythmic degradation of circadian proteins.** (a) The experimental abundance levels (solid line) and degradation rates (open circles) of the mouse PERIOD2 (PER2) protein [43]. (b) The experimental abundance levels (dots, interpolated by a solid line) and degradation rates (open circles) of PSEUDO RESPONSE REGULATOR 7 (PRR7) protein in *Arabidopsis thaliana* [44,45,48]. Horizontal white and black segments correspond to light and dark intervals, respectively. (c) A simulated protein degradation rate from the full kinetic model and its ETS- and QSSA-based estimates, when the degradation depends on a single PTM. In addition, the protein abundance profile is presented here (gray solid line). A vertical dashed line corresponds to the peak time of $-A'(t)/A(t)$ where $A(t)$ is a protein abundance. The parameters are provided in Table J in S1 Appendix. (d) The probability distribution of the peak-time difference between a degradation rate and $-A'(t)/A(t)$ for each number of PTMs ($n$) required for the degradation. The probability distribution was obtained with randomly-sampled parameter sets in Table F in S1 Appendix. (e) The probability distribution of the relative amplitude of a simulated degradation rate (top) or its estimate in Eq (13) (bottom) for each $n$, when the relative amplitude of a protein abundance is 1. (f) The probability distribution of the ratio of the simulated to estimated relative amplitude of a degradation rate for each $n$. For more details of (a)–(f), refer to Text K in S1 Appendix.

pathophysiological conditions [7,9,21,28–30]. According to previous reports, some circadian clock proteins are not only rhythmically produced but also decompose with rhythmic degradation rates [Fig 3(A) and 3(B)] [42–46]. Recently, we have suggested that the rhythmic degradation rates of proteins with circadian production can spontaneously emerge without any explicitly time-dependent regulatory mechanism of the degradation processes [42,47]. If the rhythmic degradation rate peaks at the descending phase of the protein profile and stays relatively low elsewhere, it is supposed to save much of the biosynthetic cost in maintaining a circadian rhythm. A degradation mechanism with multiple post-translational modifications

(PTMs), such as phospho-dependent ubiquitination, may elevate the rhythmicity of this degradation rate in favor of the biosynthetic cost saving [42,45]. Can the ETS explain this inherent rhythmicity in the degradation rates of circadian proteins?

First, we constructed the kinetic model of circadian protein production and degradation without the ETS or other approximations. This model attributes a circadian production rate of the protein to a circadian mRNA expression or translation rate. Yet, a protein degradation rate in the model is not based on any explicitly time-dependent regulatory processes, but on constantly-maintained proteolytic mediators such as constant E3 ubiquitin ligases and kinases. In realistic situations, the protein turnover may require multiple preceding PTMs, like mono- or multisite phosphorylation and subsequent ubiquitination. Our model covers these cases, as well. The model comprises the following equations:

$$\frac{\mathrm{d}A_0(t)}{\mathrm{d}t} = g(t) - a_0 A_0(t), \tag{10}$$

$$\frac{\mathrm{d}A_i(t)}{\mathrm{d}t} = a_{i-1} A_{i-1}(t) - a_i A_i(t), \tag{11}$$

where $A_0(t)$ and $A_i(t)$ represent the concentrations of the unmodified and $i$-th modified proteins, respectively ($i = 1, 2, \cdots, n$ and $n$ is the total number of the PTMs with $n \geq 1$), $g(t)$ is the protein production rate through mRNA expression and translation, $a_i$ denotes the protein's $(i+1)$-th modification rate ($i = 0, 1, \cdots, n-1$), and $a_n$ denotes the turnover rate of the $n$-th modified protein.

Next, we apply the ETS to the PTM processes in the model for the analytical estimation of the protein degradation rate. We observed the mathematical equivalence of the PTM processes and the above-discussed TF–DNA interactions, despite their different biological contexts (Text K in S1 Appendix). This observation leads to the estimate $r_\gamma(t)$ of the protein degradation rate as

$$r_\gamma(t) \equiv \frac{a_v}{A(t)} \min\left[ \frac{a_u}{a_u + a_v} A\left( t - \frac{1}{a_u + a_v} \right), A(t) \right]$$
$$\approx \frac{a_u a_v}{a_u + a_v} \left\{ 1 - \frac{1}{a_u + a_v} \left[ \frac{1}{A(t)} \frac{\mathrm{d}A(t)}{\mathrm{d}t} \right] + \cdots \right\}, \tag{12}$$

where $A(t)$ is the total protein concentration, $a_u$ and $a_v$ are the rates of the two slowest PTM and turnover steps in the protein degradation pathway (the step of $a_u$ precedes that of $a_v$; see Text K in S1 Appendix), and the last formula is to simplify $r_\gamma(t)$ with the Taylor expansion. The use of $r_\gamma(t)$ may not satisfactorily work for the degradation depending on many preceding PTMs, but still helps to capture the core feature of the dynamics.

Strikingly, the quasi-steady state assumption does not predict a rhythmic degradation rate, as the QSSA version of Eq (12) gives rise to a constant degradation rate, $a_u a_v/(a_u+a_v)$ (Text K in S1 Appendix). In contrast, the ETS naturally accounts for the degradation rhythmicity through the effective time delay in the degradation pathway. The last formula in Eq (12) indicates that the degradation rate would be an approximately increasing function of $-A'(t)/A(t)$ and thus increase as time goes from the ascending to descending phase of the protein profile. This predicted tendency well matches the experimental data patterns in Fig 3(A) and 3(B). Fundamentally, this degradation rhythmicity roots in the unsynchronized interplay between protein translation, modification, and turnover events [42]. For example, in the case of protein ubiquitination, ubiquitin ligases with a finite binding affinity would not always capture all newly-translated substrates, and therefore a lower proportion of the substrates can be

ubiquitinated during the ascending phase of the substrate profile than during the descending phase. The degradation rate partially follows this ubiquitination pattern. Additional PTMs like phosphorylation, if required for the ubiquitination, can further retard the full substrate modification and thereby increase the degradation rhythmicity for a given substrate profile. One may expect that these effects would be enhanced with more limited ubiquitin ligases or kinases, under the condition when the substrate level shows a strong oscillation. This expectation is supported by the relative amplitude of the degradation rate estimated by Eq (12):

$$\frac{\max_t[r_\gamma(t)] - \min_t[r_\gamma(t)]}{\langle r_\gamma(t)\rangle} \approx \frac{1}{a_u + a_v}\left\{\max_t\left[\frac{1}{A(t)}\frac{\mathrm{d}A(t)}{\mathrm{d}t}\right] - \min_t\left[\frac{1}{A(t)}\frac{\mathrm{d}A(t)}{\mathrm{d}t}\right]\right\}, \qquad (13)$$

where $\langle\cdot\rangle$ denotes a time average. Here, the relative amplitude of the degradation rate is proportional to $1/(a_u+a_v)$ as well as to the amplitude of $A'(t)/A(t)$. Therefore, limited ubiquitin ligases or kinases, and strong substrate oscillations increase the rhythmicity of the degradation rate. Given a substrate profile, multiple PTMs can further enhance this degradation rhythmicity because they invite the possibility of smaller $a_u$ and $a_v$ values than expected for the case of only a single PTM. Moreover, Eq (12) predicts that the degradation rate would peak around the peak time of $-A'(t)/A(t)$.

In the example of Fig 3(C) for a single PTM case, the simulated degradation rate from the aforementioned full kinetic model exhibits the rhythmic profile in excellent agreement with the ETS-predicted profile. Notably, the peak time of the simulated degradation rate is very close to that of $-A'(t)/A(t)$ as predicted by the ETS. Indeed, the peaks of the degradation rates show only $< 1h$ time differences from the maximum $-A'(t)/A(t)$ values across most (89–99%) of the simulated conditions of single to triple PTM cases [Fig 3(D); Text L and Table F in S1 Appendix]. In addition, for each substrate profile, the simulated degradation rate tends to become more rhythmic and have a larger relative amplitude as the number of the PTMs increases [Fig 3(E)], supporting the above argument that multiple PTMs can facilitate degradation rhythmicity. The estimated relative amplitude in Eq (13) also shows this tendency for single to double PTMs, yet not clearly for triple PTMs unlike the simulated relative amplitude [Fig 3(E)]. This inaccuracy with the triple PTMs comes from the accumulated errors over multiple PTMs in our estimation, as we indicated early. Still, the estimate in Eq (13) accounts for at least the order of magnitude of the simulated relative amplitude, as the ratio of the simulated to estimated relative amplitude almost equals 1 for a single PTM case and remains to be $O(1)$ for double and triple PTM cases [Fig 3(F)].

Together, the ETS provides a useful theoretical framework of rhythmic degradation of circadian proteins, which is hardly explained by the quasi-steady state assumption.

## Parameter estimation

The use of an accurate function of variables and parameters is important for good parameter estimation by the fitting of the parameters [13,49,50]. Parameter estimation is a crucial part of pharmacokinetic–pharmacodynamic (PK–PD) analysis for drug development and clinical study design [51,52]. Yet, the MM rate law is widely deployed for PK–PD models integrated into popular simulation and statistical analysis tools. To raise a caution against the unconditional use of the quasi-steady state assumption in parameter estimation, we here compare the accuracies of the tQSSA- and ETS-based parameter estimates. Because the sQSSA-based parameter estimates have already been known as less accurate than the tQSSA-based ones [12,50] and our own analysis supports this claim (Fig H in S1 Appendix), we will henceforth skip the use of the sQSSA.

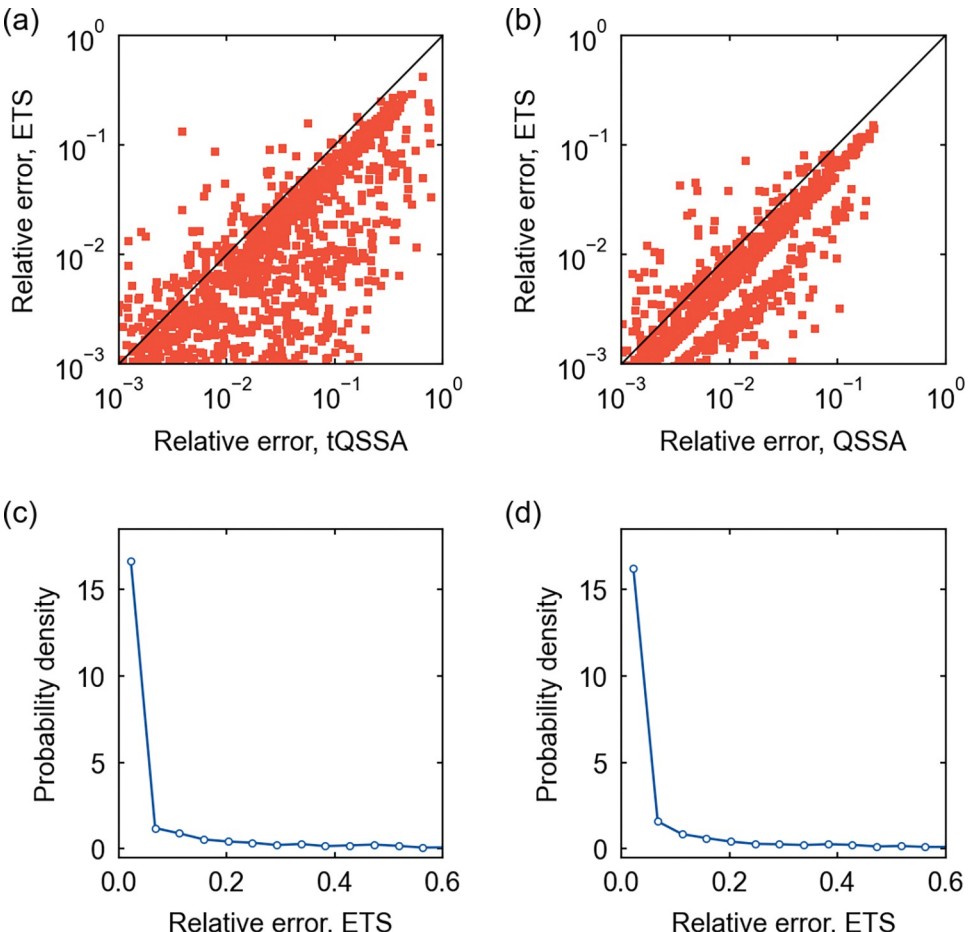

**Fig 4. Parameter estimation for protein–protein and TF–DNA interaction models.** (a) The scatter plot of the relative errors of the tQSSA- and ETS-estimated $K$ values for a protein–protein interaction model. (b) The scatter plot of the relative errors of the QSSA- and ETS-estimated $K$ values for a TF–DNA interaction model. In (a) and (b), a diagonal line corresponds to the cases where the two estimates have the same relative errors. (c) The probability distribution of the relative error of the ETS-estimated $k_\delta$ for the protein–protein interaction model in (a). (d) The probability distribution of the relative error of the ETS-estimated $k_\delta$ for the TF–DNA interaction model in (b). Regarding (a)–(d), a subset of simulated conditions gave relative errors outside the presented ranges here, but they did not alter the observed patterns. For more details, refer to Text L in S1 Appendix.

Specifically, we consider a protein–protein interaction model with time-varying protein concentrations (Text L in S1 Appendix). To the "true" profile of the protein complex [i.e., $C(t)$ in Eq (1)], we fit the ETS [$C_\gamma(t)$ in Eq (6)] or the tQSSA [$C_{tQ}(t)$ in Eq (2)] and estimate the original parameters of the model [53]: the ETS-based fitting can estimate both parameters $K$ and $k_\delta$, and the tQSSA-based fitting can estimate only $K$. Likewise, we consider a TF–DNA interaction model with time-varying TF concentration (Text L in S1 Appendix). The ETS-based fitting can estimate both $K$ and $k_\delta$, and the QSSA-based fitting can estimate only $K$.

In the case of protein–protein interactions, Fig 4(A) reveals that the ETS tends to improve the parameter estimation over the tQSSA, with more accurately estimated $K$: most of $K$ values (89.4%) estimated by the ETS show smaller relative errors than the tQSSA-based estimates and their 69.3% even show relative errors less than half the tQSSA's. In the case of TF–DNA interactions, the ETS still offers an improvement in the estimation of $K$ [Fig 4(B)]: most of the ETS-estimated $K$ values (90.3%) show smaller relative errors than the QSSA-estimated ones and their 51.8% even show relative errors less than half the QSSA's.

Unlike $K$, $k_\delta$ can only be estimated through the ETS, and hence the comparison to the tQSSA- or QSSA-based estimate is not possible. Still, $k_\delta$ is found to have the relative error $< 0.1$ for most of the ETS-based estimates, 81.3% and 81.0% in the cases of protein–protein and TF–DNA interactions, respectively [Fig 4(C) and 4(D)].

## Discussion

The quasi-steady state assumption involves the approximation by time-scale separation where the "fast" components of a system undergo instantaneous equilibrium and only the "slow" components govern the relevant dynamics. The time-scale separation has been a long practice in many different areas, such as the Monod–Wyman–Changeux model of allosteric effects, the Ackers–Johnson–Shea model of gene regulation by λ phage repressor, and the Born–Oppenheimer approximation in quantum chemistry [54–58]. If some prediction from the time-scale separation deviates from empirical data, our study may provide a useful intuition about this deviation based on an overlooked time-delay effect in that system.

We here proposed the ETS as a theoretical framework of molecular interaction kinetics with time-varying molecular concentrations. The utility of the ETS for transient or oscillatory dynamics originates in the rigorous estimation of the relaxation time in complex formation, i.e., the effective time delay. In the cases of protein–protein and TF–DNA interactions, the ETS manifests the importance of the effective time delay for the time-course molecular profiles distinct from the quasi-steady states. Accordingly, the ETS provides valuable analytical insights into the signal response time under autogenous regulation and the spontaneous establishment of the rhythmic degradation rates of circadian proteins. In addition, the ETS improves kinetic parameter estimation with a caution against the unconditional use of the quasi-steady state assumption. Our approach enhances the mathematical understanding of the time-varying behaviors of complex-complete mass-action models [38,42,59] beyond only their steady states.

Further elaboration and physical interpretation of our framework, in concert with extensive experimental profiling of molecular complexes in regulatory or signaling pathways [18,19], are warranted for the correct explanation of the interplay of cellular components and its functional consequences. Although the simulation and empirical data presented here are supportive of the ETS, direct experimental validation is clearly warranted. This validation could involve the measurement of the time-series of molecular complex concentrations, such as by mass spectrometry-based proteomics with co-immunoprecipitation, densitometry with western blotting, and enzyme-linked immunosorbent assay in the case of protein complex quantification. High temporal resolution data are preferred for their comparison with the ETS-based profiles. Lastly, comprehensive consideration of stochastic fluctuations in molecular binding events [32,60,61] beyond the TF–DNA interactions in this study would be a fruitful endeavor for more complete development of our theory, through possible extension of the existing stochastic QSSA [33,34].

## Materials and methods

The full details of theory derivation, mathematical modeling, and data sources are available in S1 Appendix. Numerical simulation and data analysis methods are presented in Text L in S1 Appendix: briefly, simulations and data analyses were performed by Python 3.7.0 or 3.7.4. Ordinary differential equations were solved by LSODA (scipy.integrate.solve_ivp) in SciPy v1.1.0 or v1.3.1 with the maximum time step of 0.05 h. Delay differential equations were solved by a modified version of the ddeint module with LSODA [62]. Splines of discrete data points were achieved with scipy.interpolate.splrep in SciPy v1.3.1. Linear regression of data points was performed with scipy.stats.linregress in SciPy v1.3.1 and then the slope of the fitted line

and $R^2$ were obtained. For the parameter selection in numerical simulations or for the null model generation in statistical significance tests, random numbers were sampled by the Mersenne Twister in random.py. To test the significance of the average of the relative errors of analytical estimates against actual simulation data, we randomized the pairing of these estimates and simulation data (while maintaining their identities as the estimates and simulation data) and measured the $P$ value (one-tailed) from the $10^4$ null configurations.

## Supporting information

**S1 Appendix.** Texts A–L, Fig A–H, and Tables A–R. **Text A.** Rate law overview and derivation. **Text B.** Amplitude overestimation with Eq (S14). **Text C.** Amplitude overestimation with simpler new rate laws. **Text D.** Rate law derivation for TF–DNA interactions. **Text E.** Preconditions of rate laws. **Text F.** Metabolic reaction and transport kinetics. **Text G.** Protein–protein interaction. **Text H.** TF–DNA interaction. **Text I.** Positive autogenous control. **Text J.** Negative autogenous control. **Text K.** Rhythmic protein degradation. **Text L**. Simulation and analysis methods. **Fig A.** Preconditions of rate laws. **Fig B.** Oxaloacetate (substrate) conversion by malate dehydrogenase (enzyme). **Fig C.** Protein–protein interaction modeling. **Fig D.** Protein ZTL–GI interaction in *Arabidopsis*. **Fig E.** TF–DNA interaction modeling. **Fig F.** Phase portrait of induction kinetics with $\eta > \eta_c$ in the case of positive autoregulation. **Fig G.** Negative autoregulation and induction kinetics. **Fig H.** The sQSSA- and tQSSA-based parameter estimation. **Table A.** Enzyme–substrate pairs of metabolic reactions in *E. coli* (refer to Text F). **Table B.** The PTS system of *E. coli* (refer to Text F). **Table C.** Parameter ranges of protein–protein and TF–DNA interaction models (refer to Texts G and H). **Table D.** Parameter ranges for ZTL profile simulation (refer to Text G). **Table E.** Parameter ranges for induction kinetics simulation [refer to Texts I and J (associated with Section *Autogenous control* in the main text)]. **Table F.** Parameter ranges for protein degradation simulation [refer to Text K (associated with Section *Rhythmic degradation of circadian proteins* in the main text)]. **Table G.** Parameters used in Fig 1(A) and 1(B). **Table H.** Parameters used in Fig 2(B)–2(D) and F. **Table I.** Simulated values of the full model-to-QSSA difference in Fig 2(D). **Table J.** Parameters used in Fig 3(C). **Table K.** Parameters used in Fig A(a) and A(b). **Table L.** Parameters used in Fig C (a). **Table M.** Parameters used in Fig C(f). **Table N.** Parameters used in Fig D(a) and D(b). **Table O.** Parameters used in Fig E(a). **Table P.** Parameters used in Fig E(d). **Table Q.** Parameters used in Fig G(b)–G(e). **Table R.** Simulated values of the QSSA-to-full model difference in Fig G(e).
(PDF)

## Acknowledgments

We thank Zhu Yang and Haneul Kim for useful discussions. This work was partially conducted with the resources of the High Performance Cluster Computing Centre, Hong Kong Baptist University, which receives funding from Research Grant Council, University Grant Committee of the HKSAR and Hong Kong Baptist University. We also acknowledge the support of the UNIST Supercomputing Center for the computing resources.

## Author Contributions

**Conceptualization:** Pan-Jun Kim.

**Data curation:** Roktaek Lim, Thomas L. P. Martin, Junghun Chae, Woo Joong Kim.

**Funding acquisition:** Cheol-Min Ghim, Pan-Jun Kim.

**Investigation:** Roktaek Lim, Thomas L. P. Martin, Junghun Chae, Cheol-Min Ghim, Pan-Jun Kim.

**Methodology:** Roktaek Lim, Thomas L. P. Martin, Junghun Chae, Cheol-Min Ghim, Pan-Jun Kim.

**Software:** Roktaek Lim, Thomas L. P. Martin, Junghun Chae.

**Supervision:** Cheol-Min Ghim, Pan-Jun Kim.

**Visualization:** Roktaek Lim, Thomas L. P. Martin, Junghun Chae.

**Writing – original draft:** Roktaek Lim, Thomas L. P. Martin, Junghun Chae, Cheol-Min Ghim, Pan-Jun Kim.

**Writing – review & editing:** Roktaek Lim, Thomas L. P. Martin, Junghun Chae, Cheol-Min Ghim, Pan-Jun Kim.

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
