## [Decision Letter · Decision Letter 0]

30 May 2023

Dear Prof. Kim,

Thank you very much for submitting your manuscript "Generalized Michaelis–Menten rate law with time‐varying molecular concentrations" (PCOMPBIOL-D-23-00559) for consideration at PLOS Computational Biology. As with all papers peer reviewed by the journal, your manuscript was reviewed by members of the editorial board and by several independent peer reviewers. Based on the reports, we regret to inform you that we will not be pursuing this manuscript for publication at PLOS Computational Biology.

The reviews are attached below this email, and we hope you will find them helpful if you decide to revise the manuscript for submission elsewhere. We are sorry that we cannot be more positive on this occasion. We very much appreciate your wish to present your work in one of PLOS's Open Access publications. 

Thank you for your support, and we hope that you will consider PLOS Computational Biology for other submissions in the future.

Sincerely,

Pedro Mendes, PhD

Section Editor

PLOS Computational Biology

Kiran Patil

Section Editor

PLOS Computational Biology

Reviewer's Responses to Questions

**Comments to the Authors: **

Reviewer #1: Attached

Reviewer #2: Review of “Generalized Michaelis—Menten rate law with time-varying molecular concentrations” by Lim et al.

The manuscript describes a novel multiscale approximation that allows for time—varying molecular concentrations, as opposed to the standard quasi-steady-state-approximation (sQSSA) and the total quasi-steady-state-approximation (tQSSA). In particular, they look at a reaction network motif of the form A + B -> C -> A+ B, where A and B are two species and C is the AB complex. They introduce an effective time delay into the steady-state concentration of the complex. The introduction of this time delay allows the authors produce more accurate approximations than both tQSSA and sQSSA. They provide several specific applications and also discuss the impact on the quality of parameter inference. 

Overall, the paper is well written. Most technical details are in the supplementary material. I did not find any technical errors. As such, I do not have any major concerns. 

Minor comment: The description/interpretation of various quasi-steady-state approximations is rather narrow/simplistic. While the description provided in “Theory overview” section is in principle okay, it is far from comprehensive. In particular, it misses out stochastic quasi-steady-state approximations and how they relate to deterministic ones.

Reviewer #3: This manuscript presents an attempt to derive a "generalized" Michaelis-Menten rate law, which according to the authors is superior than the rate laws derived using the quasi-steady-state approximation. The new rate laws is derived using a time-delay scheme.

I have substantial difficulties reading this manuscript. The work has a number of substantiative flaws in the physico-chemistry and enzymology. It is also very difficult to follow the mathematical work of the authors.

1. One of the main problems with this work is that the authors are not familiar with modern principles of theoretical enzymology. As a result of this, they introduce a number of ideas, which are fundamentally correct. For example, they assume that the quasi-steady-state approximations results from the rapid equilibrium of the complex concentration. This is fundamentally incorrect. The quasi-steady-state approximation never assumes equilibrium. It naturally arises as a result of the existence of natural scaling, which separates the reaction in two regimes: a fast regime, and a slow one. 

2. The authors claim the superiority of the total quasi-steady-state approximation over the standard quasi-steady-state approximation. The foundations of this superiority are not set in stone, but rather moving sands. Supporters of the total quasi-steady-state approximation select parameters and made numerical simulations, where the approximation shows improvements with respect to the standard quasi-steady-state approximation. Also the total quasi-steady-state approximation ha smore parameters, which make it much more difficult to implement and uniquely identify parameters. There has not been a systematic study to demonstrating than one approximation is better than the other.

3. The conditions introduced for the validity of the standard quasi-steady-state approximation - originally derived by Lee Segel - are outdated. There has been much more rigorous estimates calculated, where there is no dependency between the substrate and enzyme concentration. 

4. This reviewer doesn't understand how Eq (1) was derived, after reading Text S1. There seems to be a fundamental problem with the derivation and assumptions. Maybe I am wrong, but I couldn't follow the derivations as it is unclear how the authors have derived the total A, A(t), and total B, B(t), concentrations. They give the impression that the free A and B concentration are equal to A(t)-C(t) and B(t)-C(t). However, the reaction schemes in Figure 1 are both open. As a result of this, there is no conserve quantities. The application of the total concentrations generally requires to work with conserved reactions. As a result of this, it is unclear if the authors are applying the total-quasi-steady-state approximation well.

6. The time-delay scheme solution is very similar - structurally - to the quasi-steady-state approximations rate laws. It remains unclear the precise parameter domains, where the time-delay scheme rate law is valid. As an approximation, it must have some limitation and range of validity. The paper doesn't seem to the present one, or has a serious discussion about the validity of the new time-delay approximation.

7. Mathematical modeling with time delay comes with challenges. Delay-differential equations can have an infinity number of solutions. Parameter estimation has the same problem. To cap it all, delay differential equation numerical tools are not widely available, and require substantial expertise to be handled. It is not the typical tool used by a biochemist. This is an issue of major concern about the practical utility of the new approach.

8. It is also unclear if it is fair to compare a quasi-steady-state approximations with the time-delay approximation derived in this paper. By nature, they seems to be very different approximations, which will be valid under a different set of experimental conditions.

9. The effectiveness of the parameter estimation with the new rate law is not robust enough to determine if the new rate law is an impactful contribution to the literature. It is done in comparison with the total quasi-steady-state equation. Comparisons are limited to a restricted set of conditions, and remains unclear if it will be valid under a broader set of parameter domains.

**Have the authors made all data and (if applicable) computational code underlying the findings in their manuscript fully available?**

Reviewer #1: Yes

Reviewer #2: None

Reviewer #3: Yes

PLOS authors have the option to publish the peer review history of their article (what does this mean?). If published, this will include your full peer review and any attached files.

Reviewer #1: Yes: Jae Kyoung Kim

Reviewer #2: No

Reviewer #3: No

---

## [Decision Letter · Decision Letter 1]

21 Sep 2023

Dear Prof. Kim,

Thank you very much for submitting your manuscript "Generalized Michaelis–Menten rate law with time‐varying molecular concentrations" for consideration at PLOS Computational Biology.

As with all papers reviewed by the journal, your manuscript was reviewed by members of the editorial board and by several independent reviewers. In light of the reviews (below this email), we would like to invite the resubmission of a significantly-revised version that takes into account the reviewers' comments.

A new expert reviewer was brought in and has posed some relevant questions which you should address and modify your manuscript accordingly. (Of course, you should try to address all of the comments by all reviewers.) Any code used in this manuscript should also be made public, according to PLOS Computational Biology policies, either as supplementary material (if small enough) or in a public repository.

We cannot make any decision about publication until we have seen the revised manuscript and your response to the reviewers' comments. Your revised manuscript is also likely to be sent to reviewers for further evaluation.

Sincerely,

Pedro Mendes, PhD

Section Editor

PLOS Computational Biology

Kiran Patil

Section Editor

PLOS Computational Biology

Reviewer's Responses to Questions

**Comments to the Authors:**

Reviewer #1: I appreciate the authors’ efforts in addressing my previous comments. They have resolved all the concerns I raised, resulting in a notable enhancement in the manuscript's clarity. I recomment the manuscript for publication. I have one remaining commment.

Equations (S10) to (S12): The authors derive (S12) from (S10) by setting the integral range from -∞ to τ and omitting the term containing ¯C(τ_0). This derivation appears somewhat unclear, making it difficult to ascertain the validity of this step. I kindly request the authors to provide additional details regarding this derivation process and the underlying assumptions made.

Reviewer #2: My concerns have been sufficiently addressed.

I would like to bring to the author's attention the works of Tom Kurtz and colleagues on multiscale approximation methods that are also used to perform quasi-steady-state approximations.

1. Separation of time-scales and model reduction for stochastic reaction networks. Hye-Won Kang, and Tom Kurtz. Annals of Applied Probability.

2. Asymptotic analysis of multiscale approximations to reaction networks. Ball, Kurtz, Popovic, Rempala. Annals of Applied Probability.

3. Quasi-Steady-State Approximations Derived from the Stochastic Model of Enzyme Kinetics. Kang, KhudaBukhsh, Koeppl, and Rempala. Bulletin of Mathematical Biology.

Reviewer #3: Thank you for carefully considering my recommendations. Below you will find my comments to two points, which the authors required further clarifications.

*** Your response to Comment 2 - Part I

Yes, you are right that the expression for the complex concentration has only a single parameter in the tQSSA. However, the same can be said for the standard QSSA approximation. The fundamental problem is two-fold:

(i) In the laboratory, we can rarely observed the complex concentration; it is a short-lived chemical intermediate, particularly for steady-state kinetic experiments. My lab would like to measure complex intermediate concentrations, but we can only do this under conditions, where the complex is not anymore a short-lived intermediate, but it is the core of the reaction.

(ii) The complexity of the tQSSA lies in the total substrate concentration experiments, where there are more parameters. Additionally, it requires to measure in the laboratory the total substrate, which is not a directly observable chemical species as it requires to measure both the free substrate and intermediate complex concentration.

Your repression to data in the supplementary material shows a weak fitting overall to the tQSSA at least much more weaker of what we tend to see in the enzyme kinetics literature. This is typical for complex systems, like the tQSSA expressions.

*** Your response to Comment 2 - Part II

Your understanding of the validity of the conditions for the tQSSA is not correct. You are reading studies which are using heuristic approaches to derive the equations for the validity of the tQSSA. These approaches and their numerical solutions provide sufficient conditions for the validity of the tQSSA, but not necessary conditions, which have been proven mathematically. As such, most of the conditions published do not guarantee the validity of the approximations. The necessary conditions are much stronger. Let me bring to your attention paper 17 that you cited. In this manuscript, the necessary condition for the validity of the tQSSA is (K e0)/(Km+e0)^2 << 1. The analysis of 17 shows that tQSSA is not universally valid, as claimed by most, but only on a limiting case.

Of course, it might be possible that the reference 17 is not correct. However, the analysis in 17 seems to be more rigorous to me.

*** Your response to Comment 3

I am glad to hear that you found the references very useful. Regarding the references showing that the sQSSA is not dependent on the strate and enzyme ratio, 17 shows both that the necessary conditions for the validity of sQSSA and tQSSA are not dependent on the substrate to enzyme ratio. The abstract says "we obtain local conditions for the accuracy of standard or total quasi-steady-state. Perhaps surprisingly, our conditions do not involve initial substrate.". In my repsonse above, I provided the condition listed for the tQSSA, which is not dependent on the substrate concentration. If you read 17, or Reich and Selkov [FEBS Lett., 40 (Suppl. 1) (1974), pp. S119-S127] and justified by Palsson and Lightfoot [J. Theoret. Biol., 111 (1984), pp. 273-302], you will see that the necessary conditions for the validity of the sQSSA is e0/Km<<1.

Hopefully the authors will find these comments useful.

Reviewer #4: This manuscript focuses on the derivation of an approximate solution for bimolecular binding equation systems using the total quasi-steady state approximation (tQSSA). The approach taken focuses is to use the tQSSA as a zeroth approximation and to focus on small deviations relative to a suggested concentration scale. This has the potential benefit of qualitatively explaining the validity of the tQSSA and providing, at least in some cases, an operational model with wider validity. However, the manuscript is very hard to follow (owing to the recursive referencing of multiple supplementary sections) and the approach is somewhat cavalier (as if the authors are trying to convince themselves rather than the reviewers and general readers of the validity of their approach).

Below I detail a series of Major and Minor concerns that need be addressed for this potentially important work to be accepted.

Major Concerns

Text S1 starts out detailing clearly the approach and reveals the first key assumption S7 to correctly derive results S8-S10. As the normalized discriminant >1, inequality S7 defines a plausibly wide range for the validity of the approximation. However, from then on there is much confusion.

Firstly, the assumption S11 needs clarification and substantiation. i) Can the authors please demonstrate the time range over which this assumption is valid? This important as S12-13 give the impression that S11 is assumed to hold for all times. Can this assumption be demonstrated analytically when the sQSSA or tQSSA are valid? Ii) is S11 only assumed to hold over short durations in order to derive a local first order approximation? If so, this assumption should be made explicit and the derivation of S14 and S15 adapted accordingly. If not, please explain,

Assuming that you are planning to retain the current derivation, please explain i) why the second term in S10 does not contribute to S12, ii) and why the lower bound of the integral is -∞? Iii) why the integral in S13 ranges from 0 to infinity?

Given that S11 is assumed and can be demonstrated to be valid, can and should the derivation change and a simplified form of S10 be directly derived?

The transition from S14 to S15 needs to be justified more explicitly. Moreover, if S15 is key, can it not be derived from a reformulation of S8 in terms of c=C ®-C ®_tQ, e.g

dc/dt+(dC ®_tQ)/dt=-∆_tQ∙c

Following the derivation and validating the various assumptions made along the way is hard enough for a particular case. I would therefore ask that the authors first do so for the MM case and either move the stochastic case to a separate paper, or clearly and explicitly detail the derivation for this case, rather than just outline it. Just like the validity of the tQSSA was demonstrated gradually for different models, so should the new approximation.

Minor Concern

The figures should match the text. In particular, Figure 1A should detail the additional steps mentioned in the text (lines 120-123)

**Have the authors made all data and (if applicable) computational code underlying the findings in their manuscript fully available?**

Reviewer #1: Yes

Reviewer #2: Yes

Reviewer #3: **No: **Computer scripts or codes does not seem to be available an open repository like GitHub.

Reviewer #4: **No: **Data only appear to be provided in figure format. And I do not recall seeing a numerical methods section.

PLOS authors have the option to publish the peer review history of their article (what does this mean?). If published, this will include your full peer review and any attached files.

Reviewer #1: No

Reviewer #2: No

Reviewer #3: No

Reviewer #4: No
---

## [Decision Letter · Decision Letter 2]

24 Nov 2023

Dear Prof. Kim,

We are pleased to inform you that your manuscript 'Generalized Michaelis–Menten rate law with time‐varying molecular concentrations' has been provisionally accepted for publication in PLOS Computational Biology.

Best regards,

Pedro Mendes, PhD

Section Editor

PLOS Computational Biology

Kiran Patil

Section Editor

PLOS Computational Biology

Reviewer's Responses to Questions

**Comments to the Authors:**

Reviewer #4: The authors have adequately addressed my prior concerns.

**Have the authors made all data and (if applicable) computational code underlying the findings in their manuscript fully available?**

Reviewer #4: Yes

PLOS authors have the option to publish the peer review history of their article (what does this mean?). If published, this will include your full peer review and any attached files.

Reviewer #4: No

---

## [Editor Report · Acceptance letter]

4 Dec 2023

PCOMPBIOL-D-23-00559R2 

Generalized Michaelis–Menten rate law with time‐varying molecular concentrations

Dear Dr Kim,

I am pleased to inform you that your manuscript has been formally accepted for publication in PLOS Computational Biology. Your manuscript is now with our production department and you will be notified of the publication date in due course.

With kind regards,

Zsofi Zombor
